# Structure of the siphophage neck–Tail complex suggests that conserved tail tip proteins facilitate receptor binding and tail assembly

Hao Xiao[1,2☯], Le Tan[1☯], Zhixue Tan[1☯], Yewei Zhang[1], Wenyuan Chen[1], Xiaowu Li[3], Jingdong Song[2]*, Lingpeng Cheng[1]*, Hongrong Liu[1]*

1 Institute of Interdisciplinary Studies, Key Laboratory for Matter Microstructure and Function of Hunan Province, Key Laboratory of Low-dimensional Quantum Structures and Quantum Control, School of Physics and Electronics, Hunan Normal University, Changsha, China, 2 National Key Laboratory of Intelligent Tracking and Forecasting for Infectious Diseases, National Institute for Viral Disease Control and Prevention, Chinese Center for Disease Control and Prevention, Beijing, China, 3 School of Electronics and Information Engineering, Hunan University of Science and Engineering, Yongzhou, China

☯ These authors contributed equally to this work.
* songjd@ivdc.chinacdc.cn (JS); lingpengcheng@hunnu.edu.cn (LC); hrliu@hunnu.edu.cn (HL)

**Data Availability Statement:** The electron density maps and atomic coordinates have been deposited in the EM Data Bank and Protein Data Bank under accession codes EMD-36844, EMD-36845, EMD-

## Abstract

Siphophages have a long, flexible, and noncontractile tail that connects to the capsid through a neck. The phage tail is essential for host cell recognition and virus–host cell interactions; moreover, it serves as a channel for genome delivery during infection. However, the in situ high-resolution structure of the neck–tail complex of siphophages remains unknown. Here, we present the structure of the siphophage lambda "wild type," the most widely used, laboratory-adapted fiberless mutant. The neck–tail complex comprises a channel formed by stacked 12-fold and hexameric rings and a 3-fold symmetrical tip. The interactions among DNA and a total of 246 tail protein molecules forming the tail and neck have been characterized. Structural comparisons of the tail tips, the most diversified region across the lambda and other long-tailed phages or tail-like machines, suggest that their tail tip contains conserved domains, which facilitate tail assembly, receptor binding, cell adsorption, and DNA retaining/releasing. These domains are distributed in different tail tip proteins in different phages or tail-like machines. The side tail fibers are not required for the phage particle to orient itself vertically to the surface of the host cell during attachment.

## Introduction

Bacteriophages are the most abundant and diverse biological entities in the biosphere [1]. The majority of known phages belong to the order Caudovirales and contain a tail attached either to a portal alone or to a portal in complex with a neck (connector) located in a unique vertex of the capsid [2]. According to the tail morphology, bacteriophages are divided into 3 families: Podoviridae (short tail), Myoviridae (long contractile straight tail), and Siphoviridae (long noncontractile flexible tail) [3]. The phage tail plays key roles in host cell recognition and

36846, EMD-36847, EMD-36848, 8K35, 8K36, 8K37, 8K38 and 8K39. Underlying code for the software package used to analyse the cryo-EM data can be found in Zenodo (https://zenodo.org/records/8378566).

**Funding:** This research was supported by the National Natural Science Foundation of China (12034006 and 32071209 to H.L., 32371263 and 31971122 to L.C., 32200994 to W.C.), the Natural Science Foundation of Hunan Province, China (2020JJ2015 to X.L., 2023JJ30379 to L.C.), the Science Foundation for the State Key Laboratory for Infectious Disease Prevention and Control of China (2022SKLID203 to J.S.), and the China Postdoctoral Science Foundation (2021TQ0104 to W.C.). The funders had no role in the study design, data collection and analysis, decision to publish, or preparation of the manuscript.

**Competing interests:** The authors have declared that no competing interests exist.

**Abbreviations:** BHP, baseplate hub protein; cryo-EM, cryo-electron microscopy; dsDNA, double-stranded DNA; GTA, gene transfer agent; HD, Hub Domain; MOI, multiplicity of infection; NMR, nuclear magnetic resonance; PEG, polyethylene glycol; T6SS, type VI secretion system.

interaction. In addition, the tail serves as a channel for viral genome delivery during infection; therefore, the tail structure is an extremely interesting study subject in terms of its assembly, host recognition, and cell wall perforation mechanisms [4].

The tail structure of phages belonging to Caudovirales phage has been studied extensively. The tails of podophages and myophages are straight; therefore, they are suitable for high-resolution structural analyses. The tail structures of these phages have been reported at high resolutions. These structures include the isolated portal–tail complex of podophage T7 [5] and the entire tail of podophages T7, Pam1, sf6, and phi29 [6–9], the baseplate with 2 tail tube rings and sheath proteins of myophage T4 [10], and the tail tube of the myophage T4 [11]. However, the resolution of the entire tail structure of siphophages is difficult to improve because of its intrinsic flexibility. Some monomeric tail proteins and isolated tail complexes have been resolved from subnanometer to atomic resolutions. For example, the structures of the monomeric siphophage tail and connector proteins have been determined using nuclear magnetic resonance (NMR) spectroscopy and X-ray crystallography [12–17]. In addition, the structures of the recombinant tail tubes, tail tips, and isolated tails of siphophages have been determined by using X-ray crystallography and using cryo-electron microscopy (cryo-EM) focusing on relatively rigid regions of tail tubes, tail tips, or connectors [18–27]. The structures of a gene transfer agent (GTA, an evolutionarily related phage-like particle) [28] and type VI secretion systems (T6SS, phage tail-like bacterial nanomachines) [29,30] have also been reported. However, the structures of the neck, tail tube, and tail tip in the Siphoviridae phages remain unknown, limiting the understanding of the assembly and infection mechanisms of siphophages.

The temperate phage lambda, which infects the bacterium *Escherichia coli*, belongs to Siphoviridae. The lambda phage is used as a model system and has various therapeutic and diagnostic applications [31,32]. The virion of the lambda phage comprises an icosahedral head and a long flexible tail [2]. The head consists of a total of 415 copies of coat protein gpE, 420 copies of cementing protein gpD, 12 copies of portal protein gpB, and 48.5 kbp double-stranded DNA (dsDNA) [33]. The dodecameric portal is located at a unique icosahedral vertex and facilitates the transport of DNA in and out of the head during genome packaging and delivery [34]. The tail, which comprises the tail tube and tail tip, connects with the head through the neck. The portal, neck, and tail tube form a channel, in which the end of the dsDNA and an oligomer of tape measure protein are located [4,35].

Here, we present the cryo-EM structure of the laboratory-adapted mutant of phage lambda, commonly known as lambda "wild type." This mutant lost its host cell-binding side tail fibers as a result of laboratory adaptation [36]. Our lambda structure allows the head proteins and most proteins in the portal, neck, tail tube, and tail tip to be modeled in atomic detail. These proteins include the head proteins gpE and gpD, the portal protein gpB, the neck proteins gpW, and gpFII, the tail tube proteins gpU and $gpV_N$, and tail tip proteins gpM, gpI, gpL, gpJ, and gpH (S1 Table). Our structure shows interactions among these 246 tail protein molecules responsible for the tail assembly, receptor binding, cell adsorption, and DNA retaining/releasing. Structural comparisons between the tail tips of lambda and other long-tailed phages or tail-like machines indicate that the major domains in the tail tips are structurally conserved; however, these domains are distributed in various tail tip proteins in different phages or tail-like machines.

## Results and discussion

### Overall structure of lambda phage

Cryo-EM image shows that the mature virion of the lambda phage comprises an icosahedral head with a long and flexible tail (S1A Fig). The head structure was reconstructed at a

resolution of 3.5 Å by imposing the icosahedral symmetry (S1B and S2 Figs). The head comprises a DNA-containing capsid shell formed by the coat protein gpE and decorated by the trimers of the cementing protein gpD. Our structure of the head is essentially identical to that reported by a recent study on the lambda head [37]. We selected virion particles with a straight tail and reconstructed the entire virion structure at a resolution of approximately 20 Å (S3A and S3B Fig) using our symmetry-mismatch reconstruction method [6,38] (the software package can be downloaded from https://doi.org/10.5281/zenodo.8378566). To improve the resolution of the tail structure, we used our local refinement and reconstruction method [6,39] focusing on the portal, neck, tail tube, and tail tip. A structure of the portal–neck–tail complex at approximately 3.5 Å resolution was obtained by merging locally reconstructed structures (Fig 1). The tail, which comprises a long tail tube and a distal tail tip, attaches to the neck, which, in turn, attaches to the head (Fig 1A). The portal, neck, and tail tube form a long channel, through which a rod-like structure passes (Figs 1A, 1B and S3C). We build atomic models (S1 Table) for the portal (12-fold rings of the portal protein gpB), neck (including a 12-fold ring of the adaptor protein gpW and a hexameric ring of the stopper protein gpFII), tail tube (including a hexameric ring of the tail terminal protein gpU and 32 hexameric rings of the tail tube protein gpV), and tail tip (including a hexameric ring of the distal tail protein gpM, 3-fold trimers of the hub protein gpL, insertion protein gpI, central fiber protein gpJ, and an C-terminal α-helix of the tape measure protein gpH) (Figs 1, 2 and S4). Our mass spectrometry results (S2 Table) show that the lambda virion consists of proteins gpE, gpD, gpB, gpW, gpFII, gpU, gpV, gpM, gpH, gpL, gpI, and gpJ, as well as tail completion protein gpZ and protease protein gpC. The side tail fiber proteins (stf and tfa) and gpK were not identified using mass spectrometry. The reason for the absence of the side tail fiber proteins is that the stf gene is disrupted in the genome of lambda wild type (laboratory-adapted fiberless mutant) by frameshift mutation [36]. As for gpK, we speculated that it is a nonstructural protein that is not present in the lambda virion. Among the identified lambda proteins, gpZ and gpC were not resolved in our structures. We speculated that gpZ may lack symmetry in the tail. Indeed, gpZ homologous protein p143 in the siphophage T5 was observed to present as a monomer on the tail tip [26]. The gpC molecules might be distributed asymmetrically within the head.

## Structure of the portal and neck

The portal occupies one of the 12 vertices of the icosahedral head. It is formed by 12 copies of gpB (Figs 1A, 1B, 2A and 2B), which exhibits the canonical portal fold of phages belonging to the order Caudovirales (S5 Fig) [40]. We modeled the portal protein gpB excluding 23 N-terminal residues, 20 C-terminal residues, and a loop (residues 303–319), which were missing at the outer surface of the portal. The structure of gpB comprised 4 domains (Fig 2C): wing (residues 24–249 and 368–452), stem (residues 250–279, 343–367), clip (residues 280–342), and crown (residues 453–511). The tunnel loop, which was previously indicated to facilitate the dsDNA translocation in the portal during packaging [41] and stabilize dsDNA to avoid leakage during the phage maturation [34,42], was well resolved (Fig 2C and 2D). The asymmetric reconstruction of the portal and capsid at a resolution of 4.0 Å (S1B Fig) showed the asymmetric interactions between the gpB and gpE subunits, as observed in the structures of phages sf6 and T4 [9,43]. The portal–capsid interface was analyzed using the PISA server [44], which revealed the salt bridges between the gpB and gpE subunits (S6A and S6B Fig). Superimposition of the models of the 12 portal subunits revealed that a loop (residues 208–218) in the gpB wing domain underwent structural morphing to adapt to the portal–coat interactions (S6C Fig).

The neck comprises 2 rings composed of the adaptor and stopper proteins (Figs 1B, 2A and 2B). The adaptor ring, which is formed by 12 copies of gpW, is assembled below the portal

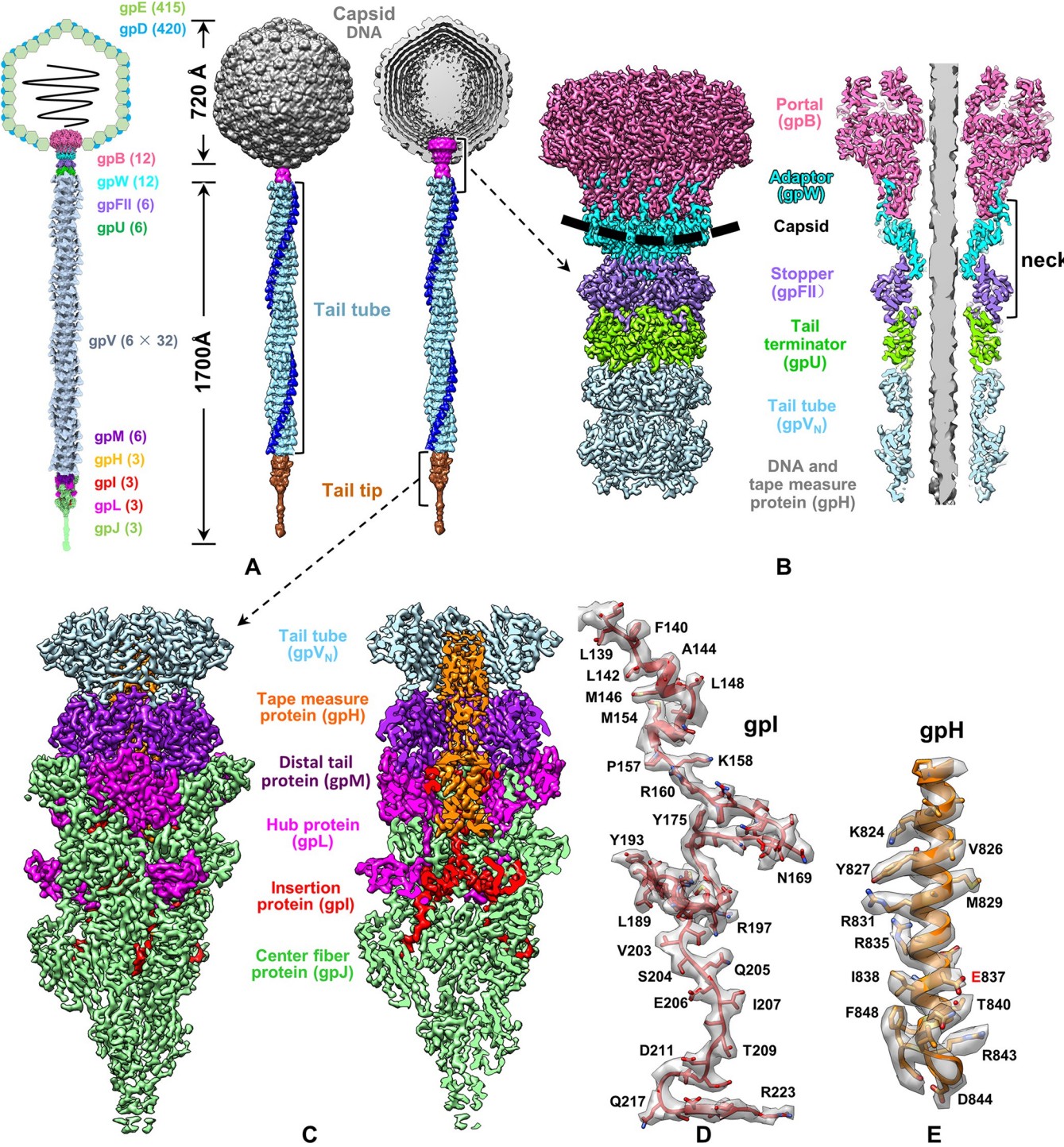

**Fig 1. Structure of the lambda virion.** (**A**) Left: scheme of the lambda virion. Each protein is shown using a different color and the numbers indicate the copy number of the proteins. Middle and right: surface and cut-open views of the lambda virion. The head, neck, tail tube, and tail tip are colored in grey, magenta, sky blue, and golden, respectively; one of the 3 protrusions of each ring of the tail tube is presented in blue to depict the right-handed helix assembly of the tail tube. (**B**) Surface and slab views of the portal (gpB), neck (gpW, and gpFII), tail terminator (gpU), and 2 tail tube (gpV$_N$) rings at a resolution of 3.5 Å. The thick dashed line marks the capsid border, and different colors have been used for different proteins. (**C**) Surface and cut-open views of the tail tip with a gpV$_N$ ring. Different colors have been used for different proteins. (**D, E**) Superposition of the resolved models of gpI and gpH on their density maps (transparent) indicates the quality of the density maps.

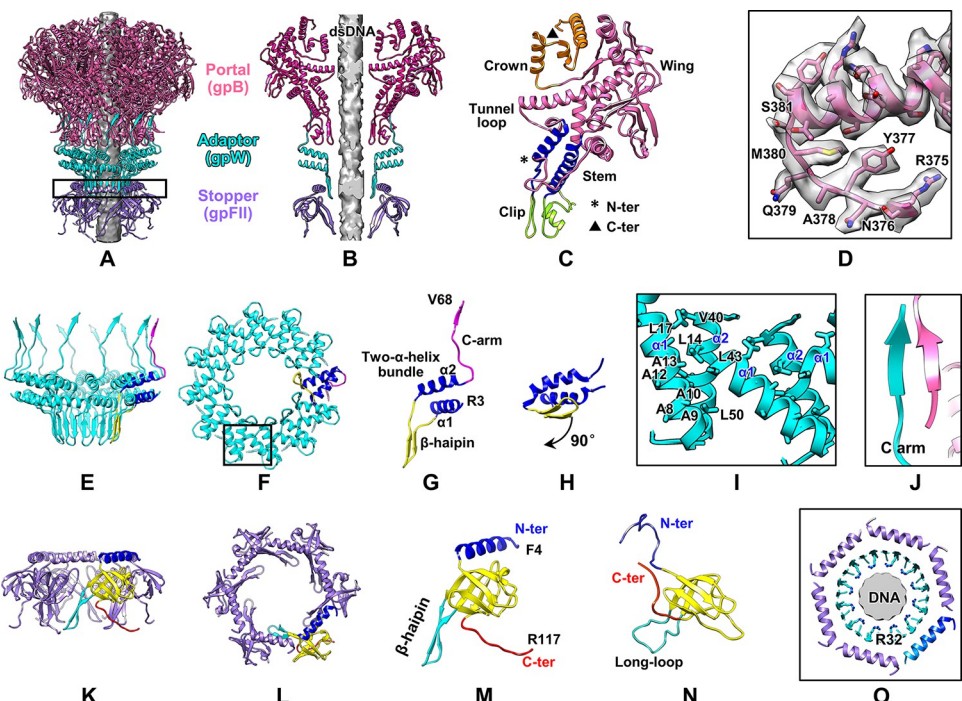

**Fig 2. Structure of the portal and neck.** (**A**, **B**) Overall and slab view of the atomic model (ribbon) of the neck. The color codes for portal (gpB), adaptor (gpW), and stopper (gpFII) are identical to those in Fig 1B. The gray rod represents the density of phage DNA. (**C**) Structure of gpB shown using various colors for its domains. (**D**) Density map (transparent) of the gpB tunnel loop superimposed on its atomic model. (**E**, **F**) Side and top views of the adaptor ring. A copy of gpW is shown using various colors for its domains. (**G**) Zoom-in view of the gpW structure in panel (**E**). (**H**) NMR structure of gpW in solution (PDB ID: 1HYW) showing that the β-hairpin domain undergoes a 90° rotation. (**I**) Zoomed-in view of the black box in panel (**F**) showing the hydrophobic interactions in the α-helix bundle domains of 2 adjacent gpW molecules. (**J**) C-arm of gpW interacts with gpB through β-sheet augmentation. (**K**, **L**) Side and top views of the stopper. One gpFII molecule is shown using various colors for its domains. (**M**) Zoomed-in view of gpFII in panel (**K**). (**N**) NMR structure of gpFII in solution (PDB ID: 1K0H). (**O**) Cross-section view of black column in panel (**A**) showing the interactions among gpFII, gpW, and DNA.

(Fig 1B). A total of 65 residues (4–68) of the 68-residue adaptor protein gpW were resolved. The structure of gpW comprises 3 domains: a β-hairpin (residues 23–36), a two-α-helix bundle (residues 4–22 and 37–54), and an extended C-arm (residues 55–68) (Figs 2E–2G and S4B). The assembly of the adaptor relies on hydrophobic interactions among the subunits of gpW (Fig 2F and 2I). The interactions of the adjacent gpW subunits are further reinforced by interactions between the adjacent β-hairpin domains, which form a β-barrel (Fig 2E). Compared with the NMR structure of a gpW monomer in solution [45], the β-hairpin in our gpW structure undergoes a 90° rotation, from parallel to vertical with respect to the two-α-helix bundle domain (Fig 2G and 2H), which facilitates the formation of the β-barrel structure. The assembly of the adaptor on the portal mainly relies on the electrostatic interactions between the electropositive adaptor top and the electronegative portal bottom (S7 Fig). In addition, the C-arm, which is absent from the NMR structure, interacts with residues 320–324 of the portal wing through β-sheet augmentation (Fig 2A and 2J).

Six copies of the stopper protein gpFII are assembled into a hexameric ring connecting the adaptor (Figs 1B, 2K and 2L). We resolved residues 4–117 of the 117-residue protein gpFII (Figs 2K–2M and S4C). The gpFII structure comprises 4 domains (Fig 2M): an N-terminal α-helix (residues 4–24), a β-sandwich (residues 25–41 and 63–107), a β-hairpin (residues 46–62)

inserted in the β-sandwich, and a C-terminal loop (residues 108–117). A comparison of our gpFII structure with the NMR structure of gpFII monomer in solution [46] revealed that the structure of gpFII monomer in solution undergoes notable conformational changes (Fig 2M and 2N) to reach the assembled state. The N-terminal loop becomes the α-helix domain. Six copies of the α-helix domains form a ring, into which the β-barrel of the adaptor rests (Fig 2A, 2B and 2O). Two other conformational changes are involved in the assembly of the tail tube ring (described below).

## Structure of the tail tube

The tail tube is located below the neck. It comprises a hexameric ring of the tail terminal protein gpU and 32 repetitively stacked hexameric rings of tail tube protein gpV (Figs 1A, 3A, 3B, S3A and S3B). The 32 rings form a right-handed helix with an axial rise of approximately 42 Å and a twist of approximately 17.5˚ for 2 adjacent rings (Fig 1A). All 131 residues of gpU comprise an antiparallel five-strand β-sheet flanked by 2 α-helices, with an O-shaped loop (O-loop, residues 25–37) inserted between β1 and β2 and an extended-loop (E-loop, residues 45–58) inserted between β2 and β3 (Figs 3C and S4D). The tail tube connects with the neck via interaction between the ring of the stopper protein gpFII and that of the tail terminal protein gpU (Fig 1A and 1B). To adapt to ring–ring interactions, the long loop (residues 46–62) observed in the structure of gpFII monomer in solution [46] becomes a β-hairpin (Fig 2M and 2N), which inserts into the inner surface of the gpU ring and interacts with the O-loop of gpU through β-sheet augmentation (Fig 3D). The C-terminal loop (residues 46–62) observed in the structure of gpFII monomer in solution [46] rotates and electrostatically interacts with the O-loop of the adjacent gpU subunit (Fig 3D). The E-loop of gpU provides an attachment site for the major tail protein gpV through electrostatic interactions (Figs 3A, 3I and S7). The surface of the gpU ring is strongly negatively charged (S7 and S8E Figs), probably facilitating the correct positioning of the packaged DNA.

Each of the 32 rings of gpV is assembled by 6 copies of the major tail protein gpV wrapped around an oligomer of the tape measure protein gpH (Fig 1B). GpV has 2 distinctive domains: the N-terminal domain (gpV$_N$, residues 3–156) and the C-terminal domain (gpV$_C$, residues 157–246). The atomic structures of the 2 individual domains in solution have been determined by NMR spectroscopy [16,17]. Our structure is similar to the lower-resolution cryo-EM structure of the lambda tail tube [22] and is essentially identical to the higher-resolution (2.7 Å) cryo-EM structure of the lambda tail tube deposited by another group in the EM Data Bank (EMD-25611) and Protein Data Bank (PDB ID: 7T2E), with an RMSD of 0.45 Å. However, our structure filtered to 6 Å resolution revealed 3 protrusions around each gpV ring; each protrusion fits well with the 2 gpV$_C$ atomic models, suggesting that each protrusion comprises 2 gpV$_C$ domains contributed by 2 neighboring gpV subunits (Figs 1A, S8A and S8B). At a higher resolution, the protrusions were poorly resolved, presumably due to their flexibility. The inner tube structure could be reconstructed to a resolution of 3.5 Å, which enabled us to build an atomic model of the gpV$_N$ domain (S1B, S4E and S8C Figs). The gpV$_N$ domain comprises a β-sandwich flanked by an α-helix. The β-sandwich is inserted by an E-loop (residues 50–78) between β3 and β4 and by an O-loop (residues 24–38) between β1 and β2 (Figs 3E and S4E). Six copies of the β-sandwich form the core of the tail tube ring (Fig 3G). The outer surface of the ring is formed by the α-helix and O-loop. Our structure of gpV$_N$ is similar to the previously reported NMR structure of gpV$_N$ [16]; however, the N-terminal, E-terminal, and the C-terminal loops are flexible in the NMR structure (Fig 3F). These loops resulted in electrostatic interactions between the adjacent gpV$_N$ rings (Figs 3H and S7) and between the gpU and gpV$_N$ rings (Figs 3I and S7). The flexibility of the tail tube may result from the interactions among

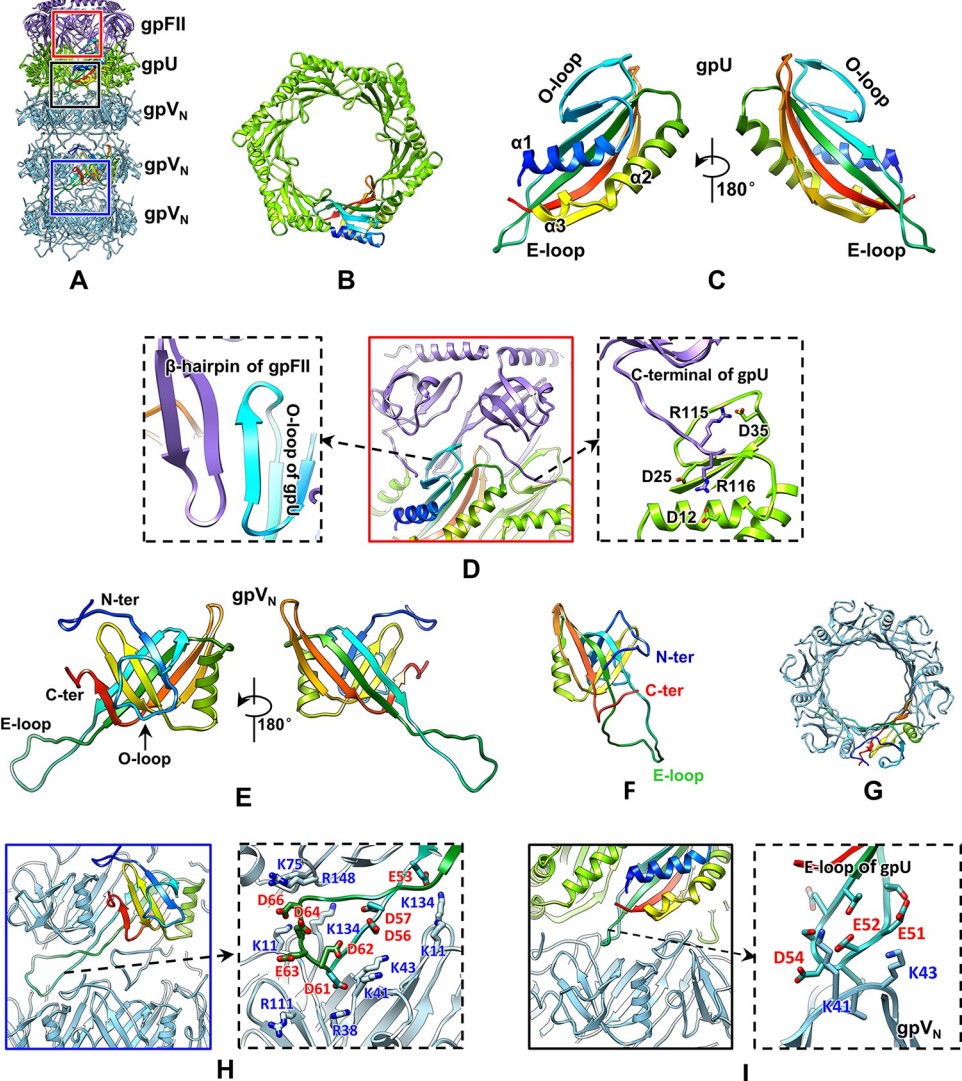

**Fig 3. Structure of the tail tube.** (**A**) Rings of gpFII, gpU, and gpV_N (only 3 rings of gpV_N are shown). The color codes are identical to those in Fig 1B, except for the fact that 1 gpU molecule and 1 gpV_N molecule are shown in rainbow colors, ranging from blue at the N-terminus to red at the C-terminus. (**B**) Top view of the gpU ring. One of the 6 gpU molecules is shown in rainbow colors. (**C**) Structure of the gpU molecule in the tail tube. (**D**) Zoomed-in view (center) of the red box in panel (**A**) showing β-sheet augmentation (left) and electrostatic (right) interactions between gpU and gpFII. (**E**) Structure of the gpV_N molecule present in the tail tube. (**F**) NMR structure of gpV_N in solution (right; PDB ID: 2k4q), showing conformational changes in the E-loop and N- and C-termini. (**G**) Top view of the hexameric gpV_N ring. One of the 6 gpU molecules is shown in rainbow colors. (**H**) Zoomed-in view (left) of the blue box in panel (**A**) showing electrostatic interactions (right) between 2 adjacent gpV_N rings. (**I**) Zoomed-in view (left panel) of the black box in panel (**A**) showing electrostatic interactions between gpU and gpV_N (right panel).

the loops. The inner surface of the gpV ring mostly bears a negative charge (S8E Fig), similar to the tail tubes of phages T5 and 80α [15,18], facilitating DNA transfer.

The architecture of gpV_N is structurally conserved across the phage tails (S9 Fig). The results of HHpred analysis indicated that the topological structure of lambda gpV_N matches those of SPP1 gp17.1, 80α gp53 and *Rhodobacter capsulatus* GTA (RcGTA) gp9 more than 99% probability (S3 Table) [18,20,28,47], although their sequence identities are low (12% to 14%).

## DNA in the neck–tail tube channel

The rod-like structure in the central channel of the neck and tail tube can be attributed to dsDNA and an oligomer of the tape measure protein gpH [4]. The genomic dsDNA of phages belonging to Caudovirales is packaged into the head through the portal channel during phage assembly [48], and the end of the last packaged dsDNA is retained in the portal [40]. In phages belonging to Caudovirales, including siphophage SPP1 [12,21,27], the narrowest part of the portal channel is formed by 12 tunnel loops. The tunnel loop, which stabilizes dsDNA in the portal tunnel in other phages [6,21], does not interact closely with the dsDNA in lambda phage (Figs 2B, S3C and S3E). In addition, the stopper ring (gpFII) of the lambda structure exhibits an open conformation (Fig 1B). This finding differs from that of previous structural studies reporting that the dsDNA end of SPP1 stops at the stopper ring (formed by gp16), at which the channel is closed by α-helices of gp16 [27]. It is intriguing that 12 arginine residues (residue 32), which are contributed by the 12 adaptor protein gpW molecules, clamp the dsDNA (Figs 1B, 2O, S3C and S3D). This finding is consistent with a previous study suggesting that gpW is required for the stabilization of lambda DNA within the head [45]. The gpV tube wraps around the gpH oligomer, which determines the length of the tail tube during assembly [49]. The gpH oligomer inside the tail tube cannot be well resolved probably because of its asymmetric assembly in the tail tube (S8D Fig).

## Structure of the tail tip complex

The tail tip complex is located at the distal end of the siphophage tail, which is the most diversified tail region. This complex is essential for host recognition and DNA ejection [18]. It comprises 6 proteins: the distal tail protein gpM, hub protein gpL, central fiber protein gpJ, insertion protein gpI, and fiber proteins stf and tfa [35]. The side tail fiber proteins stf and tfa were absent from the lambda structure because stf expression is prevented in lambda wild type by the frameshift mutation [36].

Our structure revealed that the tip is similar to an inverted cone of 410 Å in length (Figs 1C and 4A–4C). We resolved all 109 residues of the distal tail protein gpM, which has 2 conformers with minor differences in the E-loop (Figs 4D and S4F). Except that the $gpV_N$ has an extended N-terminus and C-terminus, gpM is topologically identical to $gpV_N$ (S9 Fig). GpM also has a β-sandwich, which is inserted by an E-loop and flanked by an α-helix (Fig 4D). The O-loop of $gpV_N$ is absent from gpM because of the shorter N-terminus of gpM. Structural comparisons indicate that the architecture of the gpM domain is conserved across siphophage tails (S9 Fig). Six copies of the 2 gpM conformers form a 3-fold gpM ring, which anchors the tail tip to the hexameric tail tube through the β-sandwich domain via electrostatic interactions (Figs 4B, 4E and S7). The other side of the gpM ring (E-loop) interacts with the 3-fold ring of the gpJ and gpL complex.

According to the 4 "Hub Domain" (HD) nomenclature of the siphophage and myophage baseplate hub proteins (BHPs) [4], gpJ includes the domains of HDII, HDII-insertion, HDIII, and HDIV (Fig 4G). Structural comparison between gpJ and siphophage T5 BHP pb3 [26] revealed that (i) gpJ did not have the HDI domain, and the lambda HDI domain was located in gpL (described below); (ii) the gpJ HDIV domain had an additional β-sandwich (residues 483–569) inserted between β24 (residues 477–482) and β25 (residues 570–578) (S10A and S10B Fig); (iii) the gpJ HDII-insertion domain had a shorter N-terminus compared with T5 pb3. The missing N-terminus of the gpJ HD-insertion could be compensated by the C-terminal domain of gpL (S10C and S10D Fig, described below); and (iv) an additional Ig-like domain (residues 72–215) inserted between β4 (residues 65–71) and β13 (residues 216–223) of the gpJ HDII-insertion domain (Figs 4G, S10A and S10D). The results of positive-specific

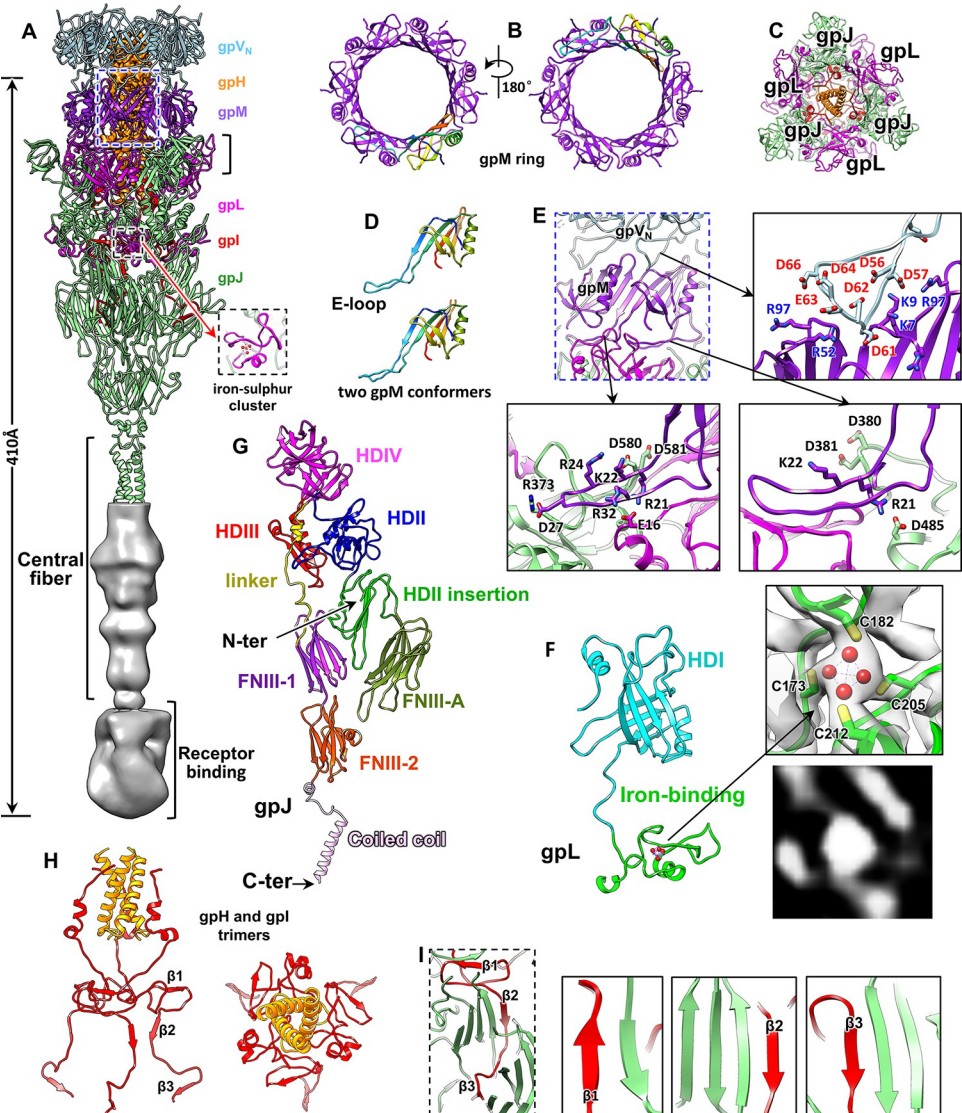

**Fig 4. Structure of the tail tip.** (**A**) Structure of the tail tip. The C-terminal region of gpJ, which was not resolved to the near-atomic resolution, is displayed using a density map (grey). (**B**) Top and bottom views of the gpM ring. One of the 6 copies of gpM is shown in rainbow colors. (**C**) Cross-section view of black column in panel (**A**) showing that the proteins gpJ and gpL form a trimeric conical tip. (**D**) Two conformers of gpM shown in rainbow colors. (**E**) Zoomed-in view (top left) of the blue box in panel (**A**) showing electrostatic interactions between gpM and gpV$_N$ (top right) and among gpM, gpJ, and gpL (bottom panels). (**F**) Structure of gpL shown using various colors for its domains (left) and the zoomed-in view (top right) of the model of the iron–sulfur cluster superimposed on its density map (transparent). A central section of the iron-binding site was shown (bottom right). (**G**) Ribbon model of gpJ shown using various colors for its domains. (**H**) Side and top views of trimeric gpI (red) and gpH (golden). (**I**) β-sheet augmentation-mediated interactions between gpI (red) and gpJ (green).

iterative basic local alignment search tool analysis revealed that the Ig-like domain belonged to the fibronectin type III domain family [50]. Thus, we designate the Ig-like domain FNIII-A (Figs 4G and S10A). Except for these differences, these gpJ domains are very similar to their counterparts in T5 [26] with RMSDs ranging from 1.5 to 3.0 Å (S11 Fig).

The four-HD scaffold is followed by 2 Ig-like domains of the fibronectin type 3 sequence family (FNIII-1 and FNIII-2) in both lambda gpJ and T5 BHP pb3. The 2 consecutive FNIIIs are connected to the HDIV through a linker domain composed of a long loop and a short α-

helix (S10A, S10B, and S11 Figs). In the T5 tail tip, the 3 copies of the 2 FNIIIs from 3 copies of pb3 are separated, which is caused by the negative patch at the surface of the pb3 FNIIIs [26]. By contrast, in the lambda tail tip, the 3 copies of the 2 FNIIIs from 3 copies of gpJ form a closed cone-like distal end of the tail tube (Fig 4A). The close interactions between these FNIIIs are enforced by a three-helix bundle below (Fig 4A). These Ig-like domains, which exist ubiquitously in phage tails, may aid in the adhesion of phage particles to bacterial cell surfaces through weakly binding to carbohydrate cell wall components such as peptidoglycan or lipopolysaccharide [51].

The protein gpL contains an N-terminal gpV$_N$-like β-sandwich domain (residues 1–166), which corresponds to the HDI domain of phage T5 pb3. Therefore, we designate the N-terminal region as the HDI domain. Three copies of the gpL HDI domains and 3 copies of gpJ HDII domains form the last ring of the tail. This ring is electrostatically stacked on the E-loops of the gpM ring (Figs 4A and S7). The E-loops of the 6 gpM molecules exhibit 2 conformations to adapt to the 3-fold symmetry of the last tail ring (Fig 4D). The C-terminal domain (residues 154–232) of gpL is topologically similar to the N-terminus of the T5 pb3 HDII-insertion domain (residues 160–209; S10E Fig), that is to say, the C-terminal region of gpL (iron-binding domain) and the N-terminal region of gpJ together correspond to the T5 pb3 HDII-insertion domain (S10A, S10C and S10D Fig).

In addition, the C-terminal domain of gpL is an iron-binding domain. A biochemical study revealed highly conserved cysteine residues coordinating an oxygen-sensitive [4Fe-4S]$^{2+}$ cluster in the C-terminal domain of gpL and homologous tip proteins of other siphopahges [52]. In our gpL structure, an additional density feature in the iron-binding domain, which was surrounded by 4 cysteine residues (Cys173, Cys182, Cys205, and Cys212) and could not be assigned to any main chain or side chain, could be modeled as the iron–sulfur cluster (Fig 4F). The iron-binding domain, which was previously hypothesized to be embedded inside the tail tip and help stabilize gpL in its assembled form [52], was found to be located at the outer surface of the tip (Fig 4A). Therefore, the functions of the iron–sulfur cluster, which is universally across gpL homologs [52], remain to be characterized.

The density in the tail tip lumen was modeled as a trimer of the gpI (residues 135–223) and a three-helix coiled coil belonging to gpH (residues 818–842; Figs 1D, 1E, 4C and 4H). The gpI trimer forms a tripod to support the coiled coil in the lumen (Fig 4C and 4H), and each gpI molecule interacts with the HDII, HDII-insertion, and FNIII-A of gpJ through 3 β-sheet augmentations (Fig 4I). The gpH coiled coil terminates the rod-like structure in the tail tip lumen (Fig 1C). A sequence-based secondary structures prediction study [53] revealed that the α-helix is located at the C-terminus of helical gpH (S12 Fig). Tape measure protein is a multifunctional protein found in both Siphoviridae and Myoviridae. During tail assembly, tape measure protein interacts with chaperone proteins to initiate tail polymerization and determines the length of the assembled tail [49,54,55]. In addition, it may transmit the host-binding signal to the phage capsid and then exit from the tail and form a channel to span the host cell envelope for phage DNA delivery [15,56,57]. Our structure suggests that the gpH oligomer is formed by gpH trimer with the gpH C-terminus located in the tail tip. The gpI trimer serves as a plug to prevent the early release of the gpH oligomer and DNA from the tail tube (Fig 4H).

## Structural comparison of gpI, gpL, and gpJ with their counterparts in other phages and phage tail-like machines

In T5, a plug domain is inserted in the HDII-insertion domain of BHP. Three copies of the plug domains also form a tripod and serve to close the tube [26] in the same location as the gpI tripod in lambda (S10B, S10D, S11A and S11B Figs). It is noteworthy that gene I is located

between genes L and J, a gene location that corresponds to the position of the T5 pb3 plug domain, except for an inserted gene K (S13 Fig). Therefore, we suggest that gpL, gpI, and the N-terminal region of gpJ (from HDII to FNIII-1 and FNIII-2) together correspond to T5 BHP pb3 (S13 Fig). The resolved N-terminal region of gpI is highly hydrophobic (residues 137-GILFSLGASMVLGGVA-152), suggesting that it might interact with the bacterial outer membrane, as does the T5 pb3 plug domain [26].

Further structural comparisons of the gpL and gpJ proteins of lambda with the homologous proteins of phage T4 [10], phage 80α [18], RcGTA [28], and T6SS [29] revealed that they all contained 4 HD domains (S11 Fig), suggesting they share a common evolutionary origin. However, T4, 80α, and T6SS lack HDII-insertion and FNIII domains. For T5, T4, 80α, and T6SS, the 4 HD domains are located within a single protein. By contrast, RcGTA has similar organization of the 4 HD domains to that of lambda. The RcGTA HDI domain is located in the N-terminal regions of gp13, while its rest 3 HD domains are located in gp15 (S11F Fig). As in lambda, the RcGTA HDII-insertion domain is also distributed in 2 tail tip proteins: the N-terminal region of the HDII-insertion domain is located in the C-terminal region of gp13, while the C-terminal region of the HDII-insertion domain is located in the N-terminal region of gp15 (S11F Fig). Indeed, the arrangement of RcGTA genes 13, 14, and 15 [28] is very similar to that of lambda genes L, K, I, and J (S13 Fig). It is noteworthy that both lambda and RcGTA have iron–sulfur clusters in the tail tip protein (gpL/gp13). Among these tail tip proteins, only lambda gpJ and T5 pb3 contain FNIII domains. The insertion/deletion and distribution of domains in these tail tips reflect the long evolutionary history of these phages and phage tail-like machines.

In addition to the aforementioned domains corresponding to T5 BHP, gpJ also contains 2 additional domains in its C-terminal region: a central fiber domain and a receptor-binding domain, which are the counterparts of the T5 central fiber protein pb4 and the receptor binding protein pb5, respectively. These 2 domains were not resolved at high resolution (Fig 4A) due to their flexibility. The flexibility of the 2 tail tip domains may enable the receptor binding domain to explore larger spaces for the bacterial receptor. We could only model the N-terminal regions of the 3 central fiber domains as three-helix bundle based on the gpJ density map (Fig 4G). The helix bundle links the central fiber and receptor binding domains to the FNIII-2 domain (Fig 4A).

We performed structure predictions for gpJ (S14A Fig) by using AlphaFold2 [58]. Comparison between the predicted gpJ structure and our gpJ structure in the tail tip showed that the domains of HDII, HDII-insertion, HDIII, and HDIV fit well (RMSD: 1.7 Å), except for the deviation of the FNIII-A, FNIII-1, and FNIII-2 domains (S14B Fig). If we segment the 3 FNIII domains from the predicted gpJ structure, they can fit well into their counterparts in our gpJ structure, with RMSDs ranging from 0.7 to 1.8 Å. The deviation of the FNIII-1 and FNIII-2 in the predicted gpJ structure was caused by the tilt of the linker domain (S14B Fig). Furthermore, the gpJ central fiber domain was predicted to be a β-strand rich region (S14A Fig). It is likely that these β-strands form β-helix in trimer, as the T5 central fiber protein pb4 [26]. The C-terminal tip of gpJ was predicted to be an Ig-like domain (S14A Fig), which could be attributed to the receptor-binding domain [59]. Through the receptor-binding domain, gpJ binds to the receptor protein LamB on the surface of the host cell [60–62]. The 2 gpJ domains were curved in the predicted structure (S14 Fig); in contrast, they appeared to be relatively straight in the density map (Fig 4A). The difference between the predicted gpJ structure and our gpJ structure might reflect the conformational change of gpJ from the monomer state in solution to the assembled state. It can be inferred that the tail tip of lambda would undergo structural changes upon receptor interaction similar to the structural changes of T5 [26]. The receptor binding process may trigger conformational changes in gpJ, which presumably disrupt the interactions among gpL, gpI, and gpJ, thus initiating the release of gpH oligomer and DNA.

### Ultrathin sections of lambda-infected cells

Ultrathin sections of the lambda-infected *E. coli* cells analyzed using electron microscopy showed that the lambda wild type particles adsorbed to the cell surface (S15 Fig), although these particles lacked side tail fibers, suggesting that the adsorption did not need the participation of the side tail fibers. This observation was in contrast to the adsorption of podophage T7, in which the 6 distal halves of the 6 side tail fibers extended vertically on the outer membrane to support the T7 particles [63]. It is probable that the conformational changes of gpL, gpI, gpJ, and the released gpH are sufficient to facilitate the lambda head and tail tube to adsorb to the cell surface.

## Materials and methods

### Bacteriophage lambda purification

*E. coli* strain MG1655 (American Type Culture Collection ID 47076) was grown in Luria–Bertani Broth (tryptone, 10 g; yeast extract, 5 g; and NaCl, 10 g/L) for 24 h at 37˚C. The cultured strain was recultured at 37˚C for 4 h. During the logarithmic growth of *E. coli*, the bacterial cells were incubated with lambda W60 phages for 4 h at 37˚C. The phage culture was lysed using chloroform. Subsequently, the cell debris was removed through centrifugation at $6,000 \times g$ for 20 min at 8˚C. The supernatant was enriched through polyethylene glycol (PEG) 8000 precipitation (10% w/v PEG in 1 M NaCl) overnight at 4˚C. The precipitated phages were resuspended in phage buffer (10 mM $MgSO_4$ and 50 mM Tris-HCl (pH 7.4)) and then purified by cesium chloride density gradient centrifugation at $90,000 \times g$ for 2 h at 8˚C. After centrifugation, bands containing phage particles were collected and dialyzed in phage buffer (10 mM $MgSO_4$ and 50 mM Tris-HCl (pH 7.4)) overnight. Final, purified infected lambda phages were in ice water for cryo-sample processing.

### Ultrathin section of lambda-adsorbed cells

The *E. coli* strain MG1655 was grown to the logarithmic phase ($OD_{600} \approx 2.0$), and the culture was collected through centrifugation at $6,000 \times g$ for 15 min at 8˚C. After the cells were resuspended in phage culture medium, purified infected lambda phages were added to the bacterial cell suspension until a multiplicity of infection (MOI) of approximately $10^5$ was achieved. The lambda phage and bacteria mixture was incubated in 37˚C water bath for 15 min and then pelleted by low speed centrifugation. The pellets were fixed with 2% paraformaldehyde–2.5% glutaraldehyde buffered in 0.1 M sodium cacodylate overnight and postfixed with 1% osmium tetroxide for 2 h. Then, the samples were dehydrated with gradient alcohol of increasing concentration until 100%. Finally, the samples were infiltrated and embedded in epoxy resin. The resin blocks were cut using ultramicrotome to get 80 nm thickness sections, which were stained with 1% uranyl acetate and 0.4% lead citrate, respectively. The ultrathin sections were examined under an FEI TF12 120 kV electron microscope at a magnification of 37,000.

### Mass spectrometry analysis

The purified lambda phage was run on a 4% to 12% SDS-PAGE, and the Coomassie-stained bands were manually excised for mass spectrometry analysis. The mass spectrometry data processing and analyses were performed using Q Exactive mass spectrometer (Thermo Scientific) and software package Proteome Discovery version 1.4. The Sequest HT search engine was used for sequence searches in the MS/MS spectra against the Uniprot database.

## Cryo-EM and data collection

An aliquot of 3 μL purified lambda phages was applied to 400-mesh Quantifoil R2/1 copper grids, which were covered with an additional 5-nm-thick layer of continuous carbon using a Q150T turbomolecular pumped coater (Quorum Technologies, United Kingdom) and glow-discharged for 20 s. The grids were loaded into a Thermo Fisher Scientific (TFS) Vitrobot (temperature, 8˚C, humidity, 100%, and blot time, 3.5 s). After blotting the excess solution on the grid 2 times, the grids were plunge frozen using liquid ethane and stored in liquid nitrogen until subsequent analyses. Cryo-EM data were recorded using the Thermo Scientific Krios G3i transmission electron microscope equipped with a Gatan imaging filter and a K3 direct electron detector, and a Cs corrector. The electron microscope was operated at 300 kV voltage, the Gatan imaging filter was used with a slit width of 20 eV to remove inelastically scattered electrons, and the K3 detector was operated in the super-resolution mode. The pixel size was calibrated using 500-nm diffraction grating replica and latex calibration standard. Data collection was done automatically using the TFS EPU software at a magnification of 53,000 (physical pixel size: 1.36 Å), which resulted in a pixel size of 0.68 Å per pixel. The accumulated dose of each movie was 32 $e^-/Å^2$, and the defocus values of the images ranged from 1.6 to 2.2 μm. Finally, 5,513 movies were collected; each movie stack comprised 32 image frames.

## Image processing

**Icosahedral reconstruction for the head.**   The defocus and astigmatism values of each cryo-EM micrograph were calculated by using GCTF [64] in RELION software [65]. A total of 97,030 particle images of the lambda head (box size 1,024 × 1,024 pixels) were boxed out from 5,513 cryo-EM images using the software ETHAN [66]. The icosahedral head of lambda was reconstructed by using our programs [67] based on the common-line algorithm [68,69].

**Symmetry-mismatch and local reconstructions.**   We manually selected 10,220 images (box size 900 × 900 pixels) of the complete lambda particles with straight tails from the cryo-EM images, which were resampled to a pixel size of 5.44 Å. Based on the orientation and center parameters of each particle image obtained from the icosahedral reconstruction, we reconstructed the intact asymmetric structure of the phage lambda using our symmetry-mismatch reconstruction method [6,38]. Briefly, for each particle image, we first localized the unique vertex (portal–neck) by searching the regions of 12 icosahedral vertices, which were determined in the icosahedral reconstruction step. Then, a low-resolution structure of the phage with the tail was obtained. The resolution was further improved through the iteration of the projection-refinement-reconstruction step. In this step, we selected one of the 60 possible icosahedral orientations for each particle image. The iteration was continued until the orientations of all particle images were stabilized and the phage structure (approximately 20 Å resolution) could not be improved further. The aforementioned steps were performed for all 97,030 particle images. Thus, we obtained a phage structure with the portal–neck and the proximal part of the tail tube but without the distal part of the tail (because the tail is flexible) at a resolution of approximately 6 Å. Next, we segmented the portal–neck from the phage structure to use it as an initial model for local refinement and reconstruction; thus, we determined the structure of the gpB (portal) and gpW (adaptor) rings at a resolution of 3.2 Å by imposing the 12-fold symmetry and the structure of the gpFII, gpU, and 2 proximal gpV rings at a resolution of 3.5 Å.

**Local reconstructions of the tail tube and neck.**   The tail tube comprises 32 repetitive hexameric rings. To increase the particle number, we manually selected 181,657 tail tube particles by using the EMAN Helix software [70] with a box size of 220 × 220 pixels. Using the low-resolution density map, which was segmented from the intact virion, as an initial model, we performed two-dimensional (2D) classification and three-dimensional (3D) reconstruction by

using RELION software [65]. A total of 121,571 particle images were selected for performing auto-refinement and 3D reconstruction with the C3 symmetry imposed. Finally, we improved resolution of the tail tube structure to 3.48 Å. The tail neck was reconstructed using the same procedure as the tail tube, except that the box size of the neck images was 400 × 400 pixels.

**Local reconstruction of the tail tip complex.** We preformed the local reconstruction of the tail tip complex by using RELION software [65] (S16 Fig). First, 70,745 tail tip particle images with a box size of 220 × 220 pixels were manually selected for 2D classification to exclude non-tip and other irrelevant segments. The classification results indicated that the centers mostly varies across particles. Using a randomly generated cylinder density as the initial model, we selected 58,215 particles from 2D classifications to perform 3D classification with the C3 symmetry. Thus, we obtained 6 types of low-resolution structures with varying center positions (S16E Fig). Furthermore, we performed Z-axis translation to correct the center of each type on the basis of the first type and reextracted tail tip images. Subsequently, 54,385 tail tip images were selected for 3D refinement with the C3 symmetry imposed. Thus, we obtained a structure of the tail tip complex at a resolution of 3.72 Å. Finally, we refined the contrast transfer function to improve the resolution of the tail tip structure to 3.44 Å.

## Model building

Using the COOT software [71], we manually built the atomic models of proteins gpB, gpW, gpFII, gpU, $gpV_N$, gpM, gpL, gpJ, gpI, and gpH (residues 818–849) on the basis of our cryo-EM density map. Furthermore, we refined the models through in real-space refinement, implemented in PHENIX [72]. The refinement and validation statistics are presented (S4 Table).

## Supporting information

**S1 Fig. Cryo-EM image and Fourier shell correlation curves.** (**A**) Cryo-EM image of mature lambda phage showing its long flexible tail. (**B**) Estimated structural resolutions of the local reconstructions of the gpB and gpW with an imposed symmetry of 12 folds (black line), the gpFII, gpU, and $gpV_N$ with an imposed symmetry of 6 folds (red line), the tail tube ($gpV_N$) with an imposed symmetry of 6 folds (blue line), the tail tip complex with an imposed symmetry of 3 folds (magenta line), and the portal–capsid without symmetry imposed (cyan line). (TIF)

**S2 Fig. Reconstruction of the icosahedral head.** (**A**) Overall view of the icosahedral head structure. Seven copies of the major capsid protein gpE arranged in an asymmetric unit are shown in purple, green, hot pink, cornflower blue, cyan, dark green, and magenta, and 6 trimers of the cementing protein gpD surrounding the asymmetric unit are shown in orange. (**B**) Zoomed-in view of the asymmetric unit in panel (**A**). The 5-, 3-, and 2-fold axes are labeled. (**C**, **D**) Density maps (grey) of the major capsid protein gpE and the trimer of the cementing protein gpD superimposed on their atomic models (ribbons). (TIF)

**S3 Fig. Symmetry-mismatch reconstruction of the intact lambda virion.** (**A**) Image of a lambda particle with a straight tail. (**B**) Overall view of an intact lambda phage at a resolution of approximately 20 Å. The numbers of the tail rings are labeled. (**C**) Slab view of the neck showing the DNA within the neck (red arrow). (**D**) Density map (transparent) at a resolution of 3.5 Å superimposed on the atomic models of gpW (cyan ribbon) and gpFII (purple ribbon), showing the interactions between the Arg32 residue of gpW and dsDNA. (**E**) Density map (transparent) at a resolution of 3.5 Å superimposed on the atomic model of gpB (hot pink

ribbon), showing the interactions between the Gln379 residue of gpB and dsDNA.
(TIF)

**S4 Fig.** Density maps (transparent or mesh) of proteins gpB (A), gpW (B), gpFII (C), gpU (D), gpV$_N$ (E), gpM (F), gpL (G), and gpJ (H) superimposed on their atomic models.
(TIF)

**S5 Fig. Comparison among the portal proteins of lambda, SPP1 (Protein Data Bank [PDB] ID: 7Z4W), T7 (PDB ID: 7BOU), and T4 (PDB ID: 3JA7).** The RMSDs between the portal proteins of lambda and SPP1, T7, and T4 are 2.52, 2.48, and 2.03 Å, respectively.
(TIF)

**S6 Fig. Portal–capsid interactions in the symmetry-mismatched binding interface.** (**A**) A schematic representation of the portal and the surrounding capsomers. The 12 portal gpB subunits (identified as 1 to 12) are in orange and cyan. The 5 M subunits (M1 to M5) and 5 N subunits (N1 to N5) of the major capsid protein gpE are in magenta and dark blue. The red arrowheads indicate salt bridges between the portal and coat subunits. (**B**) Salt bridges between the portal and coat subunits. The atomic models (ribbon) are superimposed on their density maps (transparent). (**C**) Superimposition of the 12 portal subunits revealed the structural morphing loop (residues 208–218) in gpB.
(TIF)

**S7 Fig. Electrostatic potential surfaces of the interacting regions of 2 adjacent rings.** The color scale of the electrostatic potential range is the same for all protein surfaces. The rings in the left column are oriented towards the tail tip, and the rings in the right column are oriented towards the head.
(TIF)

**S8 Fig. Density map of the tail tube.** (**A**) Two-dimensional class average of the tail tube. White, cyan, and orange arrows indicate the rings of gpV$_N$ and gpV$_C$ and the rod of gpH or dsDNA, respectively. (**B**) Density map of the tail tube filtered to a resolution of 6 Å. Superposition of 2 copies of gpV$_C$ atomic model (PDB DI: 2L04) on the protrusion (right) revealed that each protrusion contains 2 gpV$_C$ domains contributed by 2 neighboring gpV subunits. (**C**) Side and top views of the density map of the tail tube at 3.5 Å resolution. The protrusions were poorly resolved at this resolution. One monomer of gpV$_N$ is shown in yellow. (**D**) Cut-open view of the tail tube. (**E**) Electron potential on the inner channel of the neck–tail complex.
(TIF)

**S9 Fig. Structural comparison among lambda, T4, 80α, SPP1, RcGTA, and T5 in terms of the N-termini of tail tube proteins (left) and the distal tail proteins (right).** All models are shown in rainbow colors, ranging from blue at the N-termini to red at the C-termini, and the redundant parts are shown in grey.
(TIF)

**S10 Fig. Structural comparison between lambda gpJ, gpL, and T5 pb3 (baseplate hub protein).** The color code for domains is identical to that in Fig 4G. (**A**) Structure of gpJ. (**B**) Structure of T5 BHP pb3. (**C**) Structure of gpL. (**D**) Complex of a copy of gpJ, gpL, and gpI in the lambda tail tip. (**E**) Structural comparison between the C-terminal domain of the lambda gpL (residues154-232) and the N-terminus of the T5 pb3 HDII-insertion domain shows that they are topologically similar.
(TIF)

**S11 Fig. Structural comparison between lambda, other phages, and tail-like machines.** (**A-F**) Structure of the hub and central fiber proteins of lambda (gpL and gpJ), the baseplate hub or homologous proteins of phage T5 (7ZQB), phage 80α (PDB ID: 6V8I), T4 (PDB ID: 5IV5), T6SS (PDB ID: 6H3L), and RcGTA (PDB ID: 6TEH). Protein VgrG of T6SS belongs to the species *Pseudomonas aeruginosa*. The homologous domains in these proteins are in the same color. The N- and C- termini of these proteins were labeled. For the N- and C- termini of gpL and gpJ, see S10A and S10C Fig for separate views of gpJ and gpL.
(TIF)

**S12 Fig. Secondary structure of tape measure protein (gpH) predicted using the PSIPRED (version 4.0) tool.**
(TIF)

**S13 Fig. Gene arrangement of the counterpart proteins of T5, lambda, and RcGTA.** Genes encoding proteins of homologous domains are shown in the same pattern and color. The regions in white are unstructured.
(TIF)

**S14 Fig. Comparison between our gpJ structure and predicted gpJ structure.** (**A**) Structure of the gpJ structure predicted by AlphaFold. The color code for domains is identical to that in Fig 4G except that the central fiber and receptor binding domains are in pink and cornflower blue, respectively. (**B**) Comparison of our gpJ structure (cyan) and predicted gpJ structure (magenta) shows the conformational change between them.
(TIF)

**S15 Fig. Ultrathin sections images of the lambda-infected *E. coli* cells.** The scale bars represent 200 nm.
(TIF)

**S16 Fig. Tail tip reconstructed using RELION.** First, we manually selected a total of 70,745 tail tip particles (**A**, **B**) and performed 2D classification to exclude irrelevant images (**C**). Then, using a randomly generated cylinder density as the initial model (**D**), we selected a total of 58,215 particles from the 2D classifications to perform 3D classification with a C3 symmetry and obtained a total of 6 types of low-resolution structures (**E**). Third, we reextracted tail tip images (**F**) according to the centers of each type. Fourth, we performed 2D classification (**G**) and 3D auto-refinement with a C3 symmetry to obtain a density map of the tail tip complex at a resolution of 3.72 Å (**H**). Finally, we refined the contrast transfer function to improve the resolution of the tail tip structure to 3.44 Å (**I**).
(TIF)

**S1 Table. Modeled proteins of the siphophage lambda.**
(PDF)

**S2 Table. Detected proteins of the siphophage lambda by mass spectrometry.**
(PDF)

**S3 Table. Results of the HHpred analysis of the tail protein sequences of lambda phage.** For each protein, the most relevant hits are shown with the matched residues, Protein Data Bank ID, chain identifier, HHpred probability (%), E-value, and percent sequence identity in the matched regions.
(PDF)

**S4 Table. Refinement and model statistics.**
(PDF)

**S1 Appendix. The wwwPDB EM validation reports of lambda portal–capsid, portal–adaptor, neck, tail tube, and tail tip complex.**
(ZIP)

## Acknowledgments

Cryo-EM data collection was carried out at the Shuimu BioSciences Ltd. We thank Fanhao Meng, Zhenqian Guo, and other staff members at the Shuimu BioSciences for data collections.

## Author Contributions

**Conceptualization:** Jingdong Song, Lingpeng Cheng, Hongrong Liu.

**Data curation:** Hao Xiao.

**Funding acquisition:** Hongrong Liu.

**Investigation:** Xiaowu Li.

**Methodology:** Hao Xiao, Zhixue Tan, Yewei Zhang, Jingdong Song, Hongrong Liu.

**Software:** Le Tan, Wenyuan Chen.

**Validation:** Zhixue Tan, Jingdong Song.

**Visualization:** Hao Xiao, Le Tan, Zhixue Tan, Lingpeng Cheng, Hongrong Liu.

**Writing – original draft:** Hao Xiao, Le Tan, Lingpeng Cheng, Hongrong Liu.

**Writing – review & editing:** Lingpeng Cheng, Hongrong Liu.

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
