## [Editor Report · Decision Letter 0]

20 Jan 2023

Dear Dr Liu, 

Thank you for submitting your manuscript entitled "In situ structure of the connector and tail of a Siphophage" for consideration as a Research Article by PLOS Biology. Please accept my sincere apologies for the long delay in getting back to you as we consulted with an academic editor about your submission. 

Your manuscript has now been evaluated by the PLOS Biology editorial staff, as well as by an academic editor with relevant expertise, and I am writing to let you know that we would like to send your submission out for external peer review.

Once your full submission is complete, your paper will undergo a series of checks in preparation for peer review. After your manuscript has passed the checks it will be sent out for review. To provide the metadata for your submission, please Login to Editorial Manager (https://www.editorialmanager.com/pbiology) within two working days, i.e. by Jan 22 2023 11:59PM.

Kind regards,

Richard

Richard Hodge, PhD

Associate Editor, PLOS Biology

rhodge@plos.org

PLOS

---

## [Decision Letter · Decision Letter 1]

21 Mar 2023

Dear Dr Liu,

Thank you for your continued patience while your manuscript" In situ structure of the connector and tail of a Siphophage" was peer-reviewed at PLOS Biology. Please accept my sincere apologies for the delays that you have experienced during the peer review process. Your manuscript has been evaluated by the PLOS Biology editors, an Academic Editor with relevant expertise, and by three independent reviewers.

As you will see in the reviewer reports, which can be found at the end of this email, although the reviewers find the work potentially interesting, they have also raised a substantial number of important concerns. Based on their specific comments and following discussion with the Academic Editor, it is clear that a substantial amount of work would be required to meet the criteria for publication in PLOS Biology. However, given our and the reviewer interest in your study, we would be open to inviting a comprehensive revision of the study that thoroughly addresses all the reviewers' comments. Given the extent of revision that would be needed, we cannot make a decision about publication until we have seen the revised manuscript and your response to the reviewers' comments. Your revised manuscript would need to be seen by the reviewers again, but please note that we would not engage them unless their main concerns have been addressed. 

Having discussed the reviews with the Academic Editor, we think that the reporting and explanations of the methodology and results should be significantly strengthened along the lines of the reviewer’s comments, including the description of the reconstruction and the parameters used. In addition, we ask that a revised manuscript includes comparisons to the previous literature and discussions of commonalities and divergences. The reviewers also raise concerns with the lack of mechanistic insights into the infection mechanism or assembly of the lambda particle. Related to this, they note that much of the conclusions around virion assembly are inferred from previous work (Figure S11). Whilst we will not make the inclusion of new structural data essential for a revision, we do ask that you infer and report more useful mechanistic insights through careful analysis of your data (in regards to Figure S11). We also think it would be beneficial to provide additional data regarding the reconstruction and modelling in Table S3, such as providing Molprobity, Q-scores, as well as RMSD measurements for the alignments shown in Figure S5.

We appreciate that these requests represent a great deal of extra work, and we are willing to relax our standard revision time to allow you 6 months to revise your study. Please email us (plosbiology@plos.org) if you have any questions or concerns, or envision needing a (short) extension.

**IMPORTANT - SUBMITTING YOUR REVISION**

*Resubmission Checklist*

*Published Peer Review*

*PLOS Data Policy*

*Blot and Gel Data Policy*

Sincerely,

Richard

Richard Hodge, PhD

Associate Editor, PLOS Biology

rhodge@plos.org

REVIEWS:

Reviewer #1: The manuscript "In situ structure of the connector and tail of a Siphophage" by Hao Xiao et al describes the structure of the phage lambda connector-tail complex in the mature virion by cryo-electron microscopy. The composite structure was assembled from the capsid with icosahedral symmetry imposed, the portal-connecter region with C12 symmetry, the tail tube using C6 symmetry, and the tail tip with C3 symmetry. This is a sensible strategy as the flexibility of the tail tube precludes solving phage particles in toto, while the head and tail-tip that bracket the tail tube are largely rigid. The structural data presented appear to be quite comprehensive with most of the known protein components partially or fully resolved, except where there is excess flexibility such as the very tail tip and much of the tape-measure protein. The quality of density appears to be sufficient for the atomic models generated although only small regions of fitting can be shown in Figures. As the density maps are not available to the reviewer, grey-scale sections through the structures would be useful for assessing noise, artefacts and overall quality - the only section is in Fig S7a through the tail tube - and (having been burnt before by papers from other groups) I insist on seeing such data before giving a recommendation for publication. In addition, a section through the putative iron atom (lines 247-250) should show significant density due to the high scattering power of iron, and would be an addition to Figure 4F. Nonetheless, this looks overall like a structural tour-de-force, especially for detailing the complex portal-connection organization and especially the tail-tip. What it lacks is a functional payoff from this new structural data, and while something of this sort is attempted in the Supplemental Material, it is inadequate and a better attempt should be added at the end of the main text. If space is a consideration, I suggest shortening the structural details by focusing on the more significant findings in order to allow several paragraphs of conclusions. This is not a recommendation for new experimental data but for significant improvements in the presentation.

Specific concerns and considerations:

1. The term "connecter" is used to describe the portal gp8, "adapter" gpW, and "stopper" gpFII, but this is not correct. The portal is separate from the connector, being part of the capsid and involved in DNA packaging, while the connecter is a region that joins the portal to the tail after packaging is complete. Thus, it would be more correct to describe the major components as the capsid, portal, connector, tail tube and tail-tip, and in particular, the parts presented here are the portal, connector and tail (implicitly including the tail tube and tail-tip). This should be reflected in the title and throughout the manuscript. For example, line 77:

 The portal combines two rings of adaptor and stopper proteins to form the head-to-tail connector

This does not properly recognize the portal as a separate and critical component that functions not only in packaging and releasing DNA, but likely nucleates capsid assembly. 

2. Side-tail fibers are mentioned in passing (line 78, 229), but should be visible in the tail-tip structure, even as truncated knobs that most lambda lab strains include. Four side-tail fibers densities should be observed bound to gpL but none are noted. This is especially interesting in the context of the model that includes 4 trimers of the hub protein, gpL, which accounts for an apparent symmetry mismatch between a hexameric tail/trimeric tip to which the 4 side tail fibers bind.

3. Line 96: Our structure of the head is essentially identical to that reported by a recent study on the lambda head [32] except that our structure has higher resolution.

This difference - 3.8A vs 3.5A - is really not worthy of comment. I suggest removing "except that our structure has higher resolution" and instead point out any significant improvements with biological meaning, if any (none are described).

4. While this work is an advance on previous capsid and tail tube structures, and cites them, it doesn't provide any context about how the new structures differ from them, and in particular, what new functional insights are provided. More attention is given to comparison with existing NMR structures of monomers, or domains, which are of less relevance being unassembled in solution.

5. The argument (lines 208-214) that the dsDNA extends down the tail to the 3-4 gpV rings, where it meets the gpH tape measure polymer, is based on the internal tube density transitioning from hollow (gpH) to solid (dsDNA). However, density along a symmetry axis (the tail tube) is notoriously noisy, and one wonders if this is in fact where the portal-connect structure is joined to the separately-calculated tail tube structure. Additional evidence is needed to support this density assignment. Further, the tape-measure protein defines the length of the tail tube - how can it do this if it ends short of the tail tube itself? What then is governing the tail tube length? The other ideas raised, that the gpH-dsDNA interaction favors delivery and allows a release signal to propagate from the tail tip through the tape-measure protein, do not require DNA to spill into the tail tube to make this interaction if the gpH is as long as the tail tube. This paragraph should be re-thought or omitted.

6. Use of "Megatron" for the gpJ protein is new to me - I don't think this is in common use. The source seems to be ref [37] from 2020 which appropriated it from a character in the Transformers franchise, but the wording in this 2020 paper is unclear as to whether "Megatron" refers to one large multidomain protein, or several. Thus, I am also confused by line 242 that refers to Megatron in the context of gpL, which is not the "Megatron protein (gpJ)" labeled on Figure 1c. In addition, there is a completely different phage with this name - Mycobacterium phage Megatron - so I suggest "Megatron" should be dropped to avoid contributing further confusion.

7. There is an attempt at functional insights that derive from the new structure in the Supplementary information. In principle, this should be the conclusions of the main text so it's odd to read it here. However, its focus is on assembly when the paper has determined only the end product of assembly and has little to say about the steps to accomplish this. Thus, the assembly itself is inferred from other work, and much is speculative, vague and not novel. The order of accreting subunits isn't demonstrated beyond what is obvious - the head and tail assemble independently, packaging precedes binding of the head completion proteins (connector), and the tail binds to the head.

8. General comments.

Grammar could be improved in numerous places and some awkward phrasing re-thought - the authors should pay careful attention to what they write. Frequent repetition can also be avoided - for example, this redundancy in line 41:

 During the packaging of the genome, double-stranded DNA (dsDNA) is packaged into the head…

And an example of mis-wording, on line 75 - the vertex is not replaced, nothing is replaced, instead one vertex is *occupied* by the portal:

 One vertex of the icosahedron is replaced by a dodecameric portal,…

This makes no sense, although I understand the intent (line 240):

 Three copies of gpL and three copies of gpJ, surrounded alternately forming the core of the trimeric conical tail tip.

Reviewer #2: The MS "In situ structure of the connector and tail of a Siphophage" describes the atomic structure of bacteriophage lambda, namely the "wild type" lambda or the fiberless laboratory strain, as determined by cryo-electron microscopy. The authors report the composition of the virion, the copy number of most component proteins, and the nature of interaction between some components of the virus particle.

Despite containing a wealth of atomic-level information and contrary to the authors' statement (the last sequence of the Abstract), the MS provides little insight into the infection mechanism or assembly of the lambda particle for several reasons.

Infection mechanism.

"Wild type" (fiberless) lambda's infection process is well characterized functionally as lambda is a "model" phage for host receptor-induced DNA release. However, several processes remain poorly understood: a) how the interaction of the tail tip with the host LamB receptor opens the tube channel; b) structural transformation of the neck upon the initiation of DNA release; c) the function of side fibers and their role in the opening of the tube channel. Neither of these is studied in this paper. 

2. Assembly (and structure).

Unlike the laboratory-adapted and confusingly called "wild type" lambda, which lacks side fibers as a result of a laboratory-acquired mutation, the tail of the "real wild type" lambda carries four side fibers. These four side fibers are somehow attached to the sixfold (or threefold?) symmetric tail. How? The mystery of four side fibers remains unsolved.

The second big question is the interaction of the DNA with the tape measure protein in the tube channel. In a mature lambda particle, one end of DNA molecule is thought to extend into the tube channel from the capsid. The authors assume it is the case here and they distinguish between the DNA and gpH densities in the tube channel by the presence of a lumen in the latter (in gpH). Such interpretation lacks rigor. Has the length of the DNA fragment, which extends into the channel, been determined experimentally earlier? How does that length correlate with the interpretation of the cryoEM map reported here? One way to distinguish DNA from protein is to use "bubblegram" imaging: https://www.science.org/doi/10.1126/science.1214120 Perhaps, something like that can be attempted here?

As a final note on assembly, I would like to point out that one structure, no matter the resolution, cannot explain the process of assembly. And yet the Supplementary Information supplied with this paper does just that - is displays a schematic of particle assembly. It even shows four side fibers that weren't in the particle studied here. The process of lambda particle assembly has been delineated in many publications over the years. Various intermediates have been isolated and characterized. This study contains no new experiments that describe such complexes.

Besides the major concerns about the presentation of data and extrapolation of those results to subjects that have not been studied in this paper (as outlined above), there are many minor concerns that can be fixed by editing and rewriting. Of those, I will highlight only a few significant ones.

First, the title is confusing. The authors describe the structure of the complete "WT" lambda that is free in solution. Nothing is in situ here. N.B. that a phage particle consists of a head (or capsid, not capsid shell, line 39), a tail, a neck, and tail fibers (or, more generally, receptor-binding proteins). The tail is connected to the head not by a tail-to-head connector, but by a neck. Two neck proteins that interact with the portal - gpW and gpFII - are, in fact, head proteins because they do not bind to isolated tails. They are head maturation proteins. So, these proteins cannot be part of the "in situ" tail, they are neck proteins.

Second, the Introduction is a hodgepodge of sentences and concepts taken from different reviews and other papers. There is no flow and everything is just mixed together. Bacteriophage abundance (which is irrelevant for this paper) is immediately followed by a description of a particle structure with the words "pentameric", "icosahedral", "portal", "vertex", which are highly specialized terms only structure-versed people understand (and not all even). Then the DNA packaging is described in one sentence. Then, we learn about some mysterious conformation of the portal. Nothing about tail assembly though. Then we learn about different tail morphologies (shouldn't we have started with this?). And then, we learn that the tail is important for host recognition and attachment. Some general concepts and highly specialized terms are put together into unreadable sentences.

Third, the Results and Discussion section is all about minute details how this residue interacts with that residue. This is irrelevant for our understanding of particle assembly or infection mechanism. These sort of descriptions should be published in a journal that specializes on protein structure. I strongly suggest removing all the minutia into the supplementary information and compare global properties of lambda tail proteins with those of other phages. By global properties I mean surface properties, hydrophobicity, surface charge distribution, etc. Things that are actually important for function.

Two points regarding terminology.

1. Beta-strand chain swapping/insertion is called beta-sheet augmentation, not beta-augmentation. See here: https://pubmed.ncbi.nlm.nih.gov/16828554/

2. Near-atomic resolution describes a dataset in which at least some of the atoms are resolved. That is - the resolution limit must be higher that the length of at least some of the bonds in the studied molecule. For proteins, that would be ~1.4A. Resolutions lower than 1.4A are not near-atomic. The maps presented in this paper are great looking and appear to be easily interpretable in terms of amino acids, NOT atoms, so these maps a really far from "near-atomic". The highest resolution reported here is 3.2 A. This map contains only 8.3% (!) of the information expected in a near-atomic resolution map, which would be at 1.4 A resolution. All other maps are even further away from being "near atomic".

One big positive features of the MS is that the figures are done very well. Very good amount of labels, good placement and clarity. This is much appreciated. However, some of the more important figures are relegated to the Supplementary Information section at the expense of showing electron density of side chains and minutia of interactions that are functionally irrelevant. I think Fig. S3 should be in the main text. Fig. S5 is a good candidate for a main text figure as well. But in addition to the folds, the surfaces must be shown and their properties characterized. It's the surface that performs the function, not the spaghetti of the protein fold. Fig. S6 must be a main text figure. This is the only figure that deals with assembly as it shows how these rings are distinguished based on their electrostatic properties. Fig. S7 must be in the main text. I would vote for Fig. S9 to be in the main text as well. Fig. S11 should be published in a separate review paper. 

Reviewer #3: The manuscript entitled "In situ structure of the connector and tail of a Siphophage" by Xiao and co-authors has been submitted to the PLOS Biology journal. The authors present the structure of the Siphophage Lambda that infects the bacterial species Escherichia coli (E. coli). The structure has been obtained at resolutions varied from ~3.5 to 6 Angstroms between different components of the phage complex; a structure of the nearly complete phage was solved at 20 Angstrom resolution. The phage Lambda has been used for nearly two decades as a test object for usage verification methods of the cryo-electron microscopy, and software development. The authors have achieved some progress in the revelation of structural details allowing to trace the polypeptide chains of several phage proteins. This study provides an additional information on structural details of the Lambda phage and suggests some new ideas related to the phage structural organisation, that might be useful at analysis of other bacteriophages.

However, some issues must be addressed and comparison with the published structures should be done before the publication of this work:

The authors submitted the manuscript with rather unclear title: "In situ structure of the connector and tail of a Siphophage". It is easy to assume that the authors want their manuscript to be cited; in this case they must specify in the title which phage they have analysed. There are so many Siphoviridae phages, so readers will be interested in what was specifically new in this study. The authors did not discuss or indicated why this phage was important and should be subjected for this research, since as a test sample this phage was used for quite long time. 

There are some problems with logistic of the manuscript which is rather inconsistent. In the abstract they move from the tail to the connector-tail complex and its symmetry, then again to the tail. The structure of the head was not mentioned. The mechanism of infection was mentioned, but it was not explained and properly discussed in the manuscript. The main part of the introduction is dedicated to the classification of phages: at the beginning the authors started from the phage head and moved their discussion towards the tails tip, but still it was not explained why it was important to study Siphoviridae phages and, specifically, the Lambda phage. Results and discussion start from the head and overall organisation of the phage, but then the authors jumped to the connector, tail, tip, and details of structural organisation of the head were not provided. Apparently, a resolution of it was rather low and not comparable with the results published by G. Lander and J. Johnson in 2008 and by Wang C, Zeng J, Wang J. in 2022 (Structural basis of bacteriophage lambda capsid maturation. Structure, 2022 Apr;30(4):637-645.e3. DOI: 10.1016/j.str.2021.12.009. Resolution: 5.03). The structure of the Lambda tail at 2.7 A resolution has been reported and deposited to the EMBD database in 2022 (EMD-25611, Prokhorov NS, Yang Q, Catalano CE, Morais MC). No comparison was done with previously reported results. 

Unfortunately, the methods are not complete: the authors wrote that they have used their own software packages. No information was provided if these packages are available for other users, where are the links to these packages, which parameters were used during structural analysis, how high was percentage of used data in the last reconstructions, where are the atomic models used in the fitting come from. The quality of the structures reported suggests that the authors were not able to trace de novo polypeptide chains, or the accession codes must be provided if the authors used some homologues structures. What were the RMSDs between different portal proteins when the authors have done alignments: Figure S5, second raw. Where was the alignment done and according to which features? The authors have used for the SPP1 the atomic model of the 13-mer, why the authors did not use the 12-mer? The latest structure is of better quality, see 7Z4W (gp612 - gp1512 - gp166 complex).

It is interesting that the authors suggested that the tape measure protein and DNA are interwinding within the central channel of the tail. One can wonder on which evidences this suggestion was made. Do the authors have any biochemical evidence that the first pieces of the ejected DNA are intertwined with the tape measure protein? Does the tape measure protein is injected into the host cells? Did the authors measured the diameter of the inner tube of the tails? it seems that it is significantly smaller if one will compare the diameter of the dsDNA complexed with the tape measure proteins … Would be essential to provide assessments of these values. 

The authors have to provide the standard table where the overall information on the data collection will be given, the quality of the fittings, how the models were obtained, values of the correspondence between maps and polypeptide chains, RMSD, Molprobity scores, and results from the Ramachandran plots (for all fitted proteins).

The part of "the Cryo-EM and data collection" is very confusing, it seems that the authors are not familiar neither with the equipment they have used nor with the requirements for the publications. Typically, Quantifoil R2/1 copper grids (line 315) have the 2-nm thick continues carbon film, not 5-nm. If the authors have used 5 nm thick carbon layer, that have to be explained. and the proper number for this type of grids should be provided (see this and the other websites https://www.jenabioscience.com/about-us/news-blog/3342-cryo-em-grids-available). They have used unusual terminology "double-blotting". That must be explained what the authors have meant. The information on GIF is not complete. The microscope was equipped with a Gatan Quantum energy filter. What was the slit used at the data collection? What was the voltage in the microscope? What was a defocus range used at the data collection? The authors have to check how the magnification was consistent with the pixel size information. What was written in the MS is not consistent and does not correspond to the information given on websites of electron microscopes. Was the pixel size calibrated? If it was done, that has to be explained.

Image processing: what was a size of boxes for the processing of the complete phage images? Sizes pf the boxes should be given in pixels (not in the angstroms) with indication of the pixel sizes. It seems that at analysis of the different elements of the phage the authors have used different pixel sizes and box sizes. 

What was special in analysis of the symmetry mismatch: the authors were "using our symmetry-mismatch reconstruction method". No details were provided on specific features of this approach. The authors did not provide any details on what has been found. What kind interactions were revealed between proteins at pseudo equivalent positions? The authors did not explain how the portal proteins interact with the capsid.

In general, the authors did not provide any conclusion related to their research. What have they found? 

The authors did not provide reports from the EMDB database, where the quality of the structures and fitting of the atomic models would be given. This information is essential for the result quality assessments.

Minor comments:

The authors should show FSC curves for all reconstructions and indicate what was a resolution at 0.5 and at 0.143 levels.

Please rephrase: "The portal is surrounded by 12 copies of gpB, which exhibits the canonical portal fold of phages …". The portal is not "surrounded"… it is composed by 12 copies of the protein …

The authors did not provide the info on how symmetries of the different parts of the tail were verified. 

Portal proteins do not have a "funnel" shape.

Lines 209-215. "…the rod-like structure below the third gpV ring was hollow..." The authors have to indicate how big were the changes in the inner diameter of the tale tube, how big were changes in the densities of the tail rings and how much their diameters were changes? The sentence on lines 211-213 is not justified, since not information was provided.

Lines 219-222, The statement is not supported by any biochemical information. 

The authors write that "all data are fully available without restriction ". The data are not, available, no information on deposition was provided, no reports on the structure qualities and fittings. 

Line 293. A part of the sentence is confusing, should be rephrased: "…formed by a cascade of gpH trimer…" . Here there is a linguistical problem: of which "cascade" are the authors talking? Is it a process, or a set of structural elements?

The English has to be checked.

---

## [Decision Letter · Decision Letter 2]

19 Sep 2023

Dear Dr Liu,

Thank you for your patience while we considered your revised manuscript "Symmetry-mismatch structure of the bacteriophage lambda" for consideration as a Short Report at PLOS Biology. Please accept my sincere apologies for the delays that you have experienced during this round of the peer review process. Your revised study has now been evaluated by the PLOS Biology editors, the Academic Editor and two of the original reviewers. 

In light of the reviews, which you will find at the end of this email, we would like to invite you to revise the work to thoroughly address the reviewers' reports.

As you will see below, the reviewers agree that the revised manuscript is improved, but still raise concerns with the presentation and the overall quality of the writing. They also note that clarifications for both the methods used and comparisons to the previously published structures should be provided. Please note that the Academic Editor has also provided an annotated version of the manuscript file with specific comments that should also be addressed. Given these comments, we strongly encourage you to enlist the services of a professional editing service to improve the quality of the writing/language in the manuscript, as well as an colleague with expertise in phages to ensure accuracy of the text. 

We will then assess your revised manuscript and your response to the reviewers' and Academic Editor’s comments with our Academic Editor aiming to avoid further rounds of peer-review, although might need to consult with the reviewers, depending on the nature of the revisions.

We expect to receive your revised manuscript within 2 months. Please email us (plosbiology@plos.org) if you have any questions or concerns, or would like to request an extension.

*IMPORTANT*

I would also be grateful if you could address the following editorial and data-related requests that I have provided below (A-G):

(A) Please note that your manuscript is being considered as a Short Report at the journal. Short Reports have a maximum of 4 main figures, but the revised version of your manuscript now has 8 main figures. During this round of revision, we ask that the number of main figures in the manuscript is reduced to 4 upon resubmission. This can either be achieved by combining some of the main figures or by moving to the Supplementary Information. 

(B) We would like to suggest the following modification to the title: 

“Structure of the siphophage neck-tail complex suggests that conserved tail tip proteins facilitate receptor binding and tail assembly”

(C) You may be aware of the PLOS Data Policy, which requires that all data be made available without restriction: http://journals.plos.org/plosbiology/s/data-availability. For more information, please also see this editorial: http://dx.doi.org/10.1371/journal.pbio.1001797

-Supplementary files (e.g., excel). Please ensure that all data files are uploaded as 'Supporting Information' and are invariably referred to (in the manuscript, figure legends, and the Description field when uploading your files) using the following format verbatim: S1 Data, S2 Data, etc. Multiple panels of a single or even several figures can be included as multiple sheets in one excel file that is saved using exactly the following convention: S1_Data.xlsx (using an underscore).

-Deposition in a publicly available repository. Please also provide the accession code or a reviewer link so that we may view your data before publication. 

Table S2

(D) Thank you for depositing the structural data in the PDB and EMDB databases. However, I note that the data is currently on hold. We ask that you please make this data publicly available before publication.

(E) Please also ensure that each of the relevant figure legends in your manuscript include information on *WHERE THE UNDERLYING DATA CAN BE FOUND*, and ensure your supplemental data file/s has a legend.

(F) Per journal policy, as the code/software package that you have generated to perform the structural analysis is important to support the conclusions of your manuscript, we require that you make it available without restrictions upon publication. Please ensure that the code is deposited in a public data repository such as Zenodo and is sufficiently well documented and reusable, and that your Data Statement in the Editorial Manager submission system accurately describes where your code can be found (please also provide the DOI of the deposition).

(G) Please ensure that your Data Statement in the submission system accurately describes where your data can be found and is in final format, as it will be published as written there. 

**IMPORTANT - SUBMITTING YOUR REVISION**

*Resubmission Checklist*

*Published Peer Review*

*PLOS Data Policy*

*Blot and Gel Data Policy*

Sincerely,

Richard

Richard Hodge, PhD

rhodge@plos.org

REVIEWS:

Reviewer #2: The new version of the MS, now titled "Symmetry-mismatch structure of the bacteriophage lambda", has been significantly improved, but the quality of the delivery of information is still not great. The Abstract and Intro are still a little awkward, and the main text consists of long, information-dense paragraphs without subsections. The existing sections can be divided into subsections with subsection titles, and long paragraphs (almost all of them!) should be divided into smaller paragraphs. This is beyond the duty of a reviewer to fix things like this in a paper this long and this information rich. For this reason, only a few easy fixes and suggested edits are listed below.

I suggest changing the existing, somewhat odd-sounding title to one of the following three:

Symmetry mismatches in the bacteriophage lambda tail

Atomic structure of the bacteriophage lambda tail

Atomic structure of bacteriophage lambda

Abstract

 25  present the in situ structure of the neck-tail complex of the siphophage lambda wild type, which lacks side tail fiber. 

Consider: …of the siphophage lambda "wild type", the most widely used, laboratory-adapted fiberless mutant.

28  of 246 tail protein molecules responsible for neck-tail assembly and stabilization have been  …of 246 tail protein molecules forming the tail and neck have been

29  characterized. The DNA in the neck is clamped by the adaptor protein rather than the stopper 

 30  protein.  

We do not know what an adaptor and stopper are at this point, so this sentence is confusing. 

34 The side tail fibers are not required for the perpendicular 

35  adsorption of lambda.  

Consider: The side tail fibers are not required for the phage particle to orient itself perpendicular to the host cell surface during attachment. 

Intro

39  The majority of known phages belong to the order Caudovirales and contain a tail attached to  40  the pentameric vertex of the icosahedral capsid (head) through a portal or through a portal in  41  complex with neck (connector) proteins [2].  

Consider: The majority of known phages belong to the order Caudovirales and contain a tail attached to the portal protein located in a unique vertex of the capsid [2].

79  Here, we present the cryo-EM structure of the phage lambda, which is a laboratory- 

80  adapted strain and is called "lambda wild type" [35].  

Consider: Here, we present the cryo-EM structure of the laboratory-adapted mutant of phage lambda, commonly known as lambda "wild type". This mutant lost its host cell-binding side tail fibers as a result of laboratory adaptation [35].

Also consider replacing "strain" with "mutant" throughout when you describe various flavors of lambda. But this is your call - maybe "strain" is more appropriate, I am not sure.

Fig 4 needs work.

The legend is too brief. 

The two columns must be labeled "toward head" and "toward tail tip" or "head-facing side" and "tail tip-facing side".

The color bar showing the scale of e-potential must be given in kT/e.

It must be explicitly stated in the figure legend whether the coloring scheme - the color scale of the e-potential range - is the same for all surfaces.

Fig. 6E. The color bar showing the scale of e-potential must be given in kT/e.

Fig. 8. All proteins must be labeled with their trivial names, if available, or gene product names/numbers. PDB IDs can and do change. Proteins get renamed also, but this is less often. 

For example, the label T4 must be changed to T4 gp27. T6SS is T6SS VgrG N-terminal domain (or module). Also, in the figure legend, state which species VgrG belongs to.

The same is applicable to Fig. S6. All proteins in the figure must be labeled with their names or gene product numbers (e.g. T4 is actually T4 gp48, I assume?).

Reviewer #3: The authors have done (before the first revision) significant amount of analysis and spend a lot of energy on interpretation of the results. However, it is still rather difficult to read the revised MS. What was new and different from the previously published structures? The authors did not do the proper comparison of their own results with them. While the paper was improved in sense of terminology, but, unfortunately, the MS has to go to the next round of modifications and revised mainly in the part related to methods. 

Comparison of the tails structures and interpretation should be completed: what was RSMD between this new structure of the tail protein (gpV) and deposited to the EMBD database in 2022 (EMD-25611)? RMSD of 3A for the entire structure indicates noticeable differences (since this is an average value), where are the differences located? The authors were not the first who have obtained this structure. Please make the proper citations.

Sectioning of cells. The procedure of sample preparation for the sectioning of the cell infected by phages is not fully described. Which cells were used in this experiment? What were the concentrations of cells and phages used at sample preparations? What was the magnification? Why the authors insist that the tails (being so flexible) must be perpendicular to the surface of the host cell? It is not an essential requirement for the infectivity according to the figure S11 and in general for phages, that are very effective. 

The procedure of the ab-initio fitting is still not fully described as well. Did the authors fit proteins as monomers? If yes, how they have defined densities for one monomer? It seems that the authors did some homology modelling, if they have done matching with other phage proteins. The authors have to describe honestly what was done. 

Reference on 7Z4W coordinates was not given. See for the reference https://doi.org/10.1038/s41467-022-34999-8. The interpretation of the symmetry in the portal/neck complex is rather confusing. The authors write in the answers to reviewers that they have done reconstructions with different symmetries and the resolution obtained indicated what sort of symmetry had a part of the complex. However it is not very efficient approach, since there are links between C12 , C6 and C3 symmetries. Applying high rotational symmetries one will be able to see details of the components with high level of symmetry but the info related to the links between components of low-rank symmetry and components of high rank symmetry will be not interpretable. 

The order of figures is rather messy: the order of their numbering does not correspond to the order of referencing them in the text. Please check it.

The following comments will be labelled according to the version PBIOLOGY-D-23-00002_R2.pdb, the part with the corrections.

Lines 110-113. The phrase is confusing: "…the major domains in the tail tips are structurally conserved, although they are distributed in different tail tip proteins in different phages or tail-like machines". Please rephrase and explain it in the conclusions. The word "although" does not fit to the idea which the authors want to deliver.

The stopper does not clamp DNA of the phage, it may block the exit (abstract, line 33). Lines 160-162. "The tunnel loop, which was previously indicated to clamp dsDNA to avoid leakage during the phage maturation". The idea is not consistent with the other publications, where it was suggested and proved by biochemical experiments by mutations, that the portal complex is a driving motor, that pumps DNA into the procapsid. The tunnel loops help to move the DNA inside, they do not clamp DNA. 

Lines 248-254. The information written in these lines is not quite correct. Tunnel loops do not clamp the DNA. The stopper in SPP1 is not closed by the loops of gp16 but helices. The refence should be updated.

What is it "solid-liquid ethane" (line 486)? Something is wrong: was it slush ethane? Nowadays it really unusual to use such ethane. However, if they authors had bits of solid ethane during the grid freezing, the grids could be crumpled which would be not surprising and that will create problems at data collection.

Where is the gpD protein in the table S1? As it has been mentioned before the terminology should be corrected: not a "cement protein", but a "cementing protein", see doi: 10.1016/j.str.2008.05.016. The length of the protein is 110 aa and mol w. 29.67 kDa, it would be recommended to mention that in the tables.

Line 484. FEI Vitrobot -> should be Thermo Fisher Scientific (TFS) Vitrobot

Line 487. 300 kV Titan Krios G3i -> should be Thermo Scientific Krios G3i Cryo-TEM

Line 488 The K3 Gatan camera is not Summit, such name has only K2 camera.

Line 492. It is wrong expression:"Automatic data were …" -should be : "data collection was done automatically using the TFS EPU software …."

Line 493. For the pixel size 1.36A magnification given by the authors is wrong, it should be higher, more as 60 000,possibly much higher. Please check, what was it. 

Line 501. What was a reason to have such huge box size as 900x900 or (if the pixel size was 5.44 A ) = 4,900x4.900 A, if the overall size of the phage was ~2,400A (according to Fig 1)?

Fig 6. "… (B) Density map of the tail tube filtered to a resolution of 6 Å. Superposition of an atomic model of gpVC (PDB DI: 2L04) on the protrusion (right)" . It is confusing. It seems that the fit of the extra domain of gpV was not done in a best way (it seems to be tilted in a wrong direction)since it was not resolved. Different domains should be labelled in the figure.

---

## [Editor Report · Decision Letter 3]

26 Oct 2023

Dear Dr Liu,

Thank you for your patience while we considered your revised manuscript "Structure of the siphophage neck-tail complex suggests that conserved tail tip proteins facilitate receptor binding and tail assembly" for publication as a Short Report at PLOS Biology. This revised version of your manuscript has been evaluated by the PLOS Biology editors and the Academic Editor.

Our Academic Editor has now assessed your revision and has determined that the revised version satisfactorily addresses the remaining comments of the reviewers. However, the Academic Editor has provided some specific feedback regarding the quality of the English language in the manuscript text. The comments can be found below my signature and are labelled 'Comments from the Academic Editor'. We ask that you please pay specific attention to the issues raised, including the use of the word 'through' to describe protein-protein interactions (such as line 197 and 253 for instance) and discussing a feature becoming 'unstructured' at low resolution (line 233). In order to improve the quality of the writing, we ask that you please enlist the help of a native English speaking structural biologist to ensure accuracy of the text. Please note that we will not be able to proceed and accept your manuscript for publication until the text is accurate and precise, to avoid it being misleading for a general readership. 

In addition, thank you for addressing our editorial and data-related requests that we outlined during the previous round of revision. We note that the structural data deposited in the PDB and EMDB databases are still on hold for release, so we ask that you please make this data publicly available before publication. 

We expect to receive your revised manuscript within 1 month. 

*Published Peer Review History*

*Press*

Sincerely,

Richard

Richard Hodge, PhD

rhodge@plos.org

*COMMENTS FROM THE ACADEMIC EDITOR*

The paper seems to be much improved and two new interpretations of the results have emerged through stringent peer review. There are still issues with the text as you note - ambiguous use of the word 'through' when describing protein protein interactions. Also on line 233 they talk about a feature becoming unstructured at high-resolution. This is a poor-choice of words as unstructured means something very different from 'poorly resolved' - the consequence of averaging a feature that is not rigidly positioned relative to the larger entity. These data typically dissolve into noise when reconstructed to high resolution and sharpened with a global b-factor.

---

## [Editor Report · Decision Letter 4]

20 Nov 2023

Dear Dr Liu,

Thank you for the submission of your revised Short Reports "Structure of the siphophage neck-tail complex suggests that conserved tail tip proteins facilitate receptor binding and tail assembly" for publication in PLOS Biology. On behalf of my colleagues and the Academic Editor, David Bhella, I am pleased to say that we can in principle accept your manuscript for publication, provided you address any remaining formatting and reporting issues. These will be detailed in an email you should receive within 2-3 business days from our colleagues in the journal operations team; no action is required from you until then. Please note that we will not be able to formally accept your manuscript and schedule it for publication until you have completed any requested changes.

PRESS

Best wishes, 

Richard

Richard Hodge, PhD

rhodge@plos.org

PLOS
